# Potential Prognostic Role of Protein Kinase D Isoforms in Head and Neck Cancers

**DOI:** 10.3390/ijms251910274

**Published:** 2024-09-24

**Authors:** Bianka Gurbi, Kornél Dános, Ede Birtalan, Tibor Krenács, Borbála Kovács, László Tamás, Miklós Csala, Attila Varga

**Affiliations:** 1Department of Molecular Biology, Semmelweis University, 1094 Budapest, Hungary; gurbi.bianka@semmelweis.hu (B.G.); kovacs.borbala@med.semmelweis-univ.hu (B.K.); varga.attila3@semmelweis.hu (A.V.); 2Department of Oto-Rhino-Laryngology and Head and Neck Surgery, Semmelweis University, 1083 Budapest, Hungary; danos.kornel@semmelweis.hu (K.D.); birtalan.ede@semmelweis.hu (E.B.); tamas.laszlo@semmelweis.hu (L.T.); 3Department of Pathology and Experimental Cancer Research, Semmelweis University, 1085 Budapest, Hungary; krenacs.tibor@semmelweis.hu

**Keywords:** protein kinase D, head and neck cancer, prognostic marker, immunohistochemistry

## Abstract

Head and neck squamous cell carcinomas (HNSCC) are among the most common malignancies in men worldwide. Nevertheless, their clinical management is hampered by the limited availability of reliable predictive and prognostic biomarkers. Protein kinase D (PKD) isoforms contribute to major cellular processes. However, their potential role in HNSCC has not been studied systematically, which is the focus of this study. A total of 63 therapy-naive patients with squamous cell carcinoma were consecutively enrolled. Tissue microarray duplicate cores from each case were tested in situ for PKD1, PKD2, and PKD3 expression using immunohistochemistry, and the results were correlated with clinicopathological parameters. We found a high frequency of PKD1/PKD2 positive cases in oropharyngeal and PKD2 positive cases in laryngeal localizations. Only high PKD2 levels were statistically linked to elevated tumor grades, more advanced TNM (3–4) tumor stages, and p16^INK4a^ expression, while elevated PKD3 levels were associated with favorable disease-specific survival. Both PKD2 and PKD3 have been proposed to promote tumor cell proliferation, migration/invasion, and angiogenesis. However, the role of PKD3 was elusive in some cancers. Our findings suggest that testing for PKD isotypes with immunohistochemistry may support the diagnostic estimation of tumor progression and prognosis in HNSCC with a potential therapeutic relevance.

## 1. Introduction

Head and neck cancers develop from the organs located between the clavicle and the skull base: nasal cavity, oral cavity, salivary glands, pharynx, and larynx. Ninety percent of these tumors are squamous cell carcinomas (SCC) [1]. The main causative factors in the development of HNSCC are alcohol consumption, exposure to environmental pollutants, chewing of areca nut products, smoking, and human papillomavirus (HPV) infection [2,3,4]. The latter accounts for more than 50% of oropharyngeal cancers. HPV infection is mostly localized in the oral cavity; however, HPV-associated cancers predominantly arise in the oropharynx, harboring a significantly better prognosis compared to HPV-negative tumors. They also respond better to non-surgical treatments, i.e., chemotherapy, radiotherapy, radio-chemotherapy, and epidermal growth factor receptor (EGFR) targeted therapy [5]. In addition, current data suggest that HPV-positive tumors might be effectively prevented by vaccination, while for HPV-negative cancers, the options for prevention and treatment are rather limited. Despite well-characterized genomic alterations and numerous clinical trials with targeted therapeutic agents, only the EGFR inhibitor cetuximab and the immune-checkpoint PD-1 (programmed cell death protein 1) inhibitors pembrolizumab and nivolumab have been approved by the FDA and EMA as molecular targeted therapies for HNSCCs. Surgery supplemented with (chemo)radiotherapy or combined chemoradiotherapy with or without salvage surgery is still the first-line treatment option for locoregionally advanced cancers. Molecular therapies are only used in recurrent/metastatic and/or platinum-resistant cases [6,7].

In clinical practice, several EGFR inhibitors are available to head and neck tumor patients, which are either monoclonal antibodies (cetuximab and panitumumab) or small molecule tyrosine kinase inhibitors (erlotinib, gefitinib, lapatinib, afatinib) [8]. EGFR antibodies improve progression-free survival in head and neck tumors but not overall survival. Erlotinib, gefitinib, and lapatinib did not have a significant effect on survival. However, the irreversible EGFR-family inhibitor afatinib may be a promising agent [9]. Predicting the success of the therapy is difficult because the expression and mutation status of the EGFR gene do not correlate with the efficacy of anti-EGFR therapies [10]. In cancer cells, generally, Ras and Raf proteins involved in the Ras/Raf/MEK/ERK signaling pathway are frequently mutated, but their inhibition is not always effective. Furthermore, they develop resistance to EGFR inhibitors. Therefore, inhibition of downstream proteins would be a considerable strategy. For this purpose, MEK inhibitors can be applied, and in recent years, several MEK inhibitors (trametinib, selumetinib, and PD-0325901) have been successfully used for the treatment of various tumor types [11,12]. According to the aforementioned and our previous study, targeting the Ras/Ras/MEK/ERK pathway by MEK inhibitors can be an effective strategy in the treatment of HNSCC patients as well [13]. EGFR also has a direct effect on the phosphatidylinositol-4,5-bisphosphate 3-kinase (PI3K)/Akt pathway. Inhibition of this pathway can also be a promising strategy, as the mutation and increased activity of PI3K catalytic subunit (PIK3CA) have been associated with resistance of tumor cells to EGFR inhibitors, and such mutation is harbored by 5–15% of head and neck cancer cases [14,15]. In summary, a better understanding of the molecular background of HNSCCs is necessary to predict the efficacy of a certain targeted therapy.

Many prognostic markers have been identified in HNSCCs, i.e., connexin 43 (Cx43), p16^INK4a^, hepatocyte growth factor receptor (c-Met), and PIK3CA. Expression of the gap junction protein Cx43 shows a significant correlation with the survival of head and neck tumor patients, i.e., a higher expression of the protein is associated with a better prognosis [16]. Furthermore, a high level of the tumor suppressor and cell cycle regulator p16^INK4a^, which inhibits cyclin-dependent kinases (CDK) in the early G1 phase, positively correlates with better prognosis [17]. p16^INK4a^ is also used as a marker of HPV infection, but it can also be expressed in HPV-negative tumors, so the overproduction of p16^INK4a^ does not necessarily indicate the presence of HPV [10]. In contrast, the mutation and overexpression of the tyrosine kinase receptor c-Met or the PIK3CA correlate with poorer patient survival [18,19].

The protein kinase D (PKD) family of isoenzymes PKD1, PKD2, and PKD3 have been implicated in the regulation of diverse cellular processes, including cell growth, proliferation, cell migration/invasion, apoptosis, and epithelial-mesenchymal transition (EMT) [20]. Based on their structural homology, PKDs are serine/threonine kinases of the calcium/calmodulin-dependent kinase superfamily [21,22,23], which can be activated by hormones, growth factors, diacylglycerol, and oxidative stress [20]. In line with their key roles in cell fate, PKDs are also involved in cancer development and progression-related pathways in a tissue type and PKD isoform-dependent manner, either as oncogenes or tumor suppressors [24]. PKD1 protein was frequently downregulated in HNSCC cell lines and human clinical surgical samples. The role of PKD1 in cell proliferation was found to be cell context-dependent, as neither ectopic expression nor depletion of PKD1 altered HNSCC cell proliferation. However, the growth rate of HNSCC xenografts was enhanced by the induction of ectopic PKD1 expression [25]. In another study, elevated PKD3 levels were correlated with the incidence of metastasis and poor prognosis in oral squamous cell carcinoma (OSCC), a subtype of HNSCC [26]. Furthermore, it has been suggested that PKD3 may enhance the malignant progression of OSCC through modulation of the transcription factor KLF16. However, the potential prognostic role of PKD isotypes in the same HNSCC cohort had not been studied systematically.

In this work, the expression of PKD1, PKD2, and PKD3 isoforms at the protein level was tested in situ in a series of HNSCCs from different anatomical regions using a sensitive immunohistochemistry protocol and the results were correlated with the clinicopathological parameters of the tumors, including localization, tumor grade, TNM (tumor, node, metastasis) stage and survival.

## 2. Results

Following immunohistochemical staining, 54 samples for PKD1, 55 for PKD2, and 58 for PKD3 were evaluable for analysis. Typical tissue samples with low or high PKD isoform levels are shown in Figure 1.

High PKD1 expression was found in 29/54 (54%) of the HNSCC samples. PKD1 status did not correlate with sex (*p* = 0.90), smoking (*p* = 0.68), alcohol consumption (*p* = 0.75), tumor size (*p* = 0.52), lymph-node metastasis (*p* = 0.21), distant metastasis (*p* = 0.37), p16^INK4a^ status (*p* = 0.26), HPV infection (*p* = 0.49) or PKD3 status (*p* = 0.36).

Of the 55 tumor samples, 41 (75%) cases showed high PKD2 levels. PKD2 status did not correlate with sex (*p* = 0.07), smoking (*p* = 0.76), alcohol consumption (*p* = 0.61), tumor size (*p* = 0.47), lymph-node metastasis (*p* = 0.26), distant metastasis (*p* = 0.12), HPV infection (*p* = 1.00) or PKD3 status (*p* = 0.61).

High PKD3 expression was found in 31/58 (53%) of HNSCCs. PKD3 status did not correlate with sex (*p* = 0.31), smoking (*p* = 0.90), alcohol consumption (*p* = 0.66), tumor size (*p* = 0.52), lymph-node metastasis (*p* = 0.21), distant metastasis (*p* = 0.37), p16^INK4a^ status (*p* = 0.61) or HPV infection (*p* = 0.43).

We compared the PKD isoform expression profiles in tumors of different localization and found a significant correlation for PKD1 (*p* = 0.028) and PKD2 (*p* = 0.002).

High PKD1 protein levels were found to be markedly more prevalent in tissue samples derived from the oropharynx (26% vs. 7%), while low PKD1 protein levels were more frequent in hypopharyngeal tumors (24% vs. 11%). Tissue samples from laryngeal tumors showed a rather even balance between low and high PKD1 expression (15% vs. 17%, respectively) (Figure 2A).

High PKD2 protein expression was predominant both in oropharyngeal (31% vs. 2%) and laryngeal (27% vs. 4%) tumors, while low and high expressions were nearly equal in the hypopharyngeal tumor samples (20% vs. 16%, respectively) (Figure 2B).

In contrast to these isoforms, PDK3 protein levels did not show considerable differences in any tumor groups of distinct anatomical localization, i.e., the frequency of low and high expression was nearly the same in all topologies (*p* = 0.929) (Figure 2C).

The possible association between the expression of PKD isoforms was analyzed in all tissue samples irrespective of tumor localization. A significant correlation was found between PKD1 and 2 profiles (*p* = 0.012), as samples with high PKD1 protein levels were predominantly also high in PKD2 expression (47%) (Figure 3). A similar correlation was not detected in samples with low PKD1 protein levels or between other isoform combinations.

Tissue samples from tumors of all locations were also tested for correlation between PKD isoform expression and the presence or absence of the known HNSCC prognostic biomarker p16^INK4a^. We found that all p16^INK4a^ positive samples were among those characterized by high PKD2 expression levels (*p* = 0.041) (Figure 4).

We compared the PKD isoform expression profiles in tumors of different grades and found a significant correlation for PKD2 (*p* = 0.008). High expression of PKD2 was strongly associated with poor tumor grade: 40% of grade 2 samples and 33% of grade 3 samples belonged to the high PKD2 category, while most of grade 1 samples showed low PKD2 protein levels (Figure 5B). Although the low PKD1 profile was more frequent in grade 1 and the high PKD1 profile dominated in grade 3, these apparent differences were not statistically significant (*p* = 0.093) (Figure 5A). No relationship was found between PKD3 expression and tumor grade, i.e., the distribution of low and high PKD3 profiles was almost even among samples from tumors of different grades (*p* = 0.881) (Figure 5C).

PKD isoform expression profiles were also compared in tumors of different TNM stages, and the results were in good agreement with the outcome of the grade-related analysis. A significant correlation was only seen for PKD2 (*p* = 0.03). High PKD2 levels were detected more frequently in samples with advanced tumor progression, indicating that PKD2 protein expression was increased in stages 3 and 4 (Figure 6B). As for the PKD1 isoform, we found a tendency of elevated expression with tumor progression, but the differences were not significant (*p* = 0.051) (Figure 6A). The low or high expression levels of PKD3 were again evenly distributed and revealed no relationship with the TNM stage of the tissue samples (*p* = 0.159) (Figure 6C).

The observed correlations between PKD protein levels and tumor grade or TNM stage suggested that certain isoform profiles may have prognostic value. Therefore, we compared the disease-specific survival associated with high or low expression of each isoform. Although the differences were not significant, the results showed that tumors with high expression of either PKD1 or PKD2 had a markedly worse prognosis than the corresponding low-expression groups (Figure 7A,B). Interestingly, despite the above-described lack of correlation of the studied parameters with PKD3 protein levels, we found that tumors with a high PKD3 expression profile showed a significantly better prognosis than those with low PKD3 levels (*p* = 0.008) (Figure 7C).

## 3. Discussion

Head and neck cancers are among the most common types of cancer in men worldwide [1]. Despite well-characterized genomic alterations and numerous clinical trials, the treatment options are still rather limited [6,7]. Therefore, studies searching for further cancer development and progression-related prognostic markers are needed to improve the prediction of disease outcomes.

The expression profile and exact role of PKD isoforms have not been systematically studied in head and neck cancers. Nevertheless, reduced levels of PKD1 and, in metastasis, increased levels of PKD3 have been reported in HNSCC compared to normal tissue [25,26]. Here, we investigated the in situ expression of all three PKD isoforms in the same cohort of HNSCC locoregional subtypes for the first time, and we revealed a remarkable differential expression of some isoforms within some regions. We found a dominantly high expression of both PKD1 and PKD2 in the oropharynx, while samples from the hypopharynx showed dominantly low PKD1 levels and nearly equal proportions of high and low PKD2 levels. In contrast, in samples from the larynx, a dominantly high expression of PKD2 and the same frequency of high and low levels of PKD1 were found. Interestingly, we did not see any significant difference in the frequency of PKD3 expression within the locoregional subgroups of HNSCC samples.

We also analyzed the relationship between the protein levels of the three PKD isoforms and found that samples with elevated PKD1 expression also showed predominantly high PKD2 levels. Our results are in line with the published functional similarities of these PKD isotypes, particularly concerning their involvement in the regulation of the cell cycle at the G2 and mitotic phases and cell proliferation through the Raf-MEK-Erk signaling pathway [27,28]. The observed association deserves a closer investigation in HNSCC cell models.

p16^INK4a^ is involved in cell cycle regulation as a major cyclin-dependent kinase (CDK) inhibitor in the early G1 phase [29]. Its anti-proliferative effect is based on the inhibition of the cyclin D1–CDK4/6 complex, which activates the cell cycle by hyperphosphorylating pRb (retinoblastoma protein). In HPV-infected cells, an oncogenic viral gene product, E7 protein, exerts its transforming effect essentially by mimicking the CDK-mediated hyperphosphorylation of pRb, which leads to increased expression of p16^INK4a^ [30]. However, this tumor suppressor can also be expressed in HPV-negative tumors, suggesting that the overproduction of p16^INK4a^ does not necessarily indicate the presence of HPV [17]. In this study, we only detected p16^INK4a^ expression in samples with high PKD2 expression. This phenomenon may be due to the positive role of PKD2 in cell cycle regulation or to HPV infection itself, which also requires further experiments [28].

PKD2 is also known to participate in tumor cell migration and invasion. It promotes tumor progression, epithelial-to-mesenchymal transition (EMT), and is associated with more malignant cancer phenotypes [31]. Previous studies have shown that knockdown of PKD2 markedly inhibited breast cancer cell migration and invasion [32]. Knockdown of PKD2 in lung adenocarcinoma cell lines significantly reduced the expression of mesenchymal markers (N-cadherin, vimentin) and transcription factors (Twist, Snail) that promote EMT; it also inhibited cell migration and invasion [33]. It has also been demonstrated that PKD2 mediates cell invasion through the expression and secretion of MMP-1 in glioblastoma and MMP-7 and MMP-9 in pancreatic cancer. PKD2 expression was found to be upregulated in hepatocellular carcinoma (HCC) and shown to enhance TNF-induced EMT and invasion of HCC cells [34]. In this study, we investigated tumor grade and TNM stage, which reflect tumor dedifferentiation and metastatic features, respectively. We found that high PKD2 levels were detected more frequently in samples of elevated tumor grades and poorly differentiated tumor cells, which results may be consistent with the contribution of PKD2 to EMT of HNSCC. In our tested HNSCC cohort, PKD2 protein levels were also significantly higher in advanced metastatic TNM categories (stages 3 and 4) than in lower stages. These results are also in line with the observations that suggest an association between elevated PKD2 levels and malignant cancer phenotypes [35]. Therefore, our findings allow us to conclude that PKD2 may be a potential marker of HNSCC metastasis/tumor progression.

In many tumor types, PKD2 mediates EMT, tumor cell migration, and invasion in cooperation with PKD3, which has also been confirmed to participate in HNSCC tumor cell migration. Furthermore, the level of PKD3 was suggested to be elevated in metastatic head and neck tumor tissue samples [26]. However, our study showed no significant correlation between tumor metastatic pathological parameters and PKD3 protein levels in HNSCC. Also, while high PKD3 expression was linked by others to poor overall survival (OS) of HNSCC patients, our results showed that elevated PKD3 protein levels were associated with better disease-specific survival (DSS) [26]. The reason for this discrepancy may be that OS and DSS are different types of survival, i.e., the OS rate is used to determine the survival of patients without any information about the underlying causes of death, while the DSS rate only takes into account the mortality caused by a specific disease [36].

In summary, by detecting PKD protein isotypes in situ in HNSCC using sensitive immunohistochemistry, we found a high frequency of PKD1 and PKD2 positive cases in oropharyngeal and PKD2 positive cases in laryngeal localizations. High PKD2 levels were linked with elevated tumor grades and more advanced (3–4) TNM tumor stages, while elevated PKD3 levels were correlated with better patient outcomes. Although the presented study is limited by a relatively small sample size, we conclude that testing for PKD isotypes with immunohistochemistry may support the better diagnostic estimation of tumor progression and prognosis in HNSCC.

## 4. Materials and Methods

### 4.1. Patients

Altogether, 63 therapy-naive patients were consecutively enrolled who were diagnosed with squamous cell carcinoma of the oropharynx, hypopharynx, or larynx at the Department of Oto-Rhino-Laryngology and Head and Neck Surgery, Semmelweis University between 2012 and 2014. All subjects gave their informed consent for inclusion before they participated in the study. The study was conducted in accordance with the Declaration of Helsinki, and the protocol was approved by the Semmelweis University’s Regional, Institutional Scientific and Research Ethics Committee (ethical license No: 105/2014). The most important characteristics of our cohort are shown in Table 1.

### 4.2. Tissue Microarray (TMA) and Immunohistochemistry

TMA blocks containing 2 mm diameter duplicate cores from the formalin-fixed and paraffin-embedded (FFPE) tissue samples of each enrolled case were created using the TMA Master tool (3DHistech Ltd., Budapest, Hungary). Tissue sections cut in 4 µm thickness, mounted on adhesion slides, were stained with hematoxylin and eosin (H & E), which allowed the accurate TMA core selection from the area representative for each case.

TMA slides were immunostained using PKD subtype-specific rabbit polyclonal antibodies anti-PKD1 (HPA028834) anti-PKD3 (HPA029529) from Merck (Kenilworth, NJ, USA), and mouse polyclonal antibody anti-PKD2 (PA5–21427) from Thermo Fisher Scientific (Waltham, MA, USA) using a BenchMark Ultra (Roche-Ventana, Tucson, AZ, USA) automated instrument as described before [37]. The protocol briefly involved the following main steps: high pH antigen retrieval in CC1 buffer for 60 min; incubation with PKD1 and PKD2 primary antibodies in 1:100 dilutions for 30 min, and detection using the Ultraview system for 20 min with DAB-peroxidase visualization. Immunostaining for PKD3 was performed manually at room temperature as follows: routine dewaxing and rehydration, antigen retrieval in pH 9.0 Tris (0.1 M)-EDTA (0.01 M) buffer in an electric pressure cooker (Avair, Biofa, Veszprém, Hungary) for 20 min; incubation with PKD3 primary antibody in 1:100 dilution and then with a micropolymer detection system (Histols, Histopathology Ltd. Pecs, Hungary) both for 60 min. Immunoreactions were revealed using a DAB-hydrogen peroxide kit (Thermo Fisher Scientific, Waltham, MA, USA), resulting in brown staining.

All stained TMA slides were digitalized using a Pannoramic 1000 whole slide scanner and analyzed with eye control in the SlideViewer program (3DHistech Ltd., Budapest, Hungary) for the evaluation of PKD1, PKD2, and PKD3 immunoreactions, an alternative 4-stier H-scoring approach was used. Briefly, the percentage of stained tumor cells was multiplied by the intensity grade of the staining (grade 1: negative; grade 2: weak; grade 3: moderate; and grade 4: intense), which resulted in values between 0 and 400. Cores with scores 0 (0), 1 to 100 (1), 101 to 200 (2), 201 to 300 (3), and 301 to 400 (4) were referred to as negative, low, moderately low, moderately high or high protein expression, respectively. For statistical analysis, scores were dichotomized along different threshold values. The most reproducible threshold for all assessors was set up when scores 0 and 1 were considered as low/negative expression, whereas scores 2, 3, and 4 were taken as high/positive expression (Table 2).

Due to the cutting of the TMA blocks, the digestion of the tissues, and other factors, some of the samples did not retain enough tumor cells for testing. These cases represent the “no data” category. Out of the possible 63 samples, 54 samples could be evaluated for PKD1, 55 for PKD2, and 58 for PKD3.

### 4.3. Statistical Analysis

Statistical correlations between PKD expression patterns and patient data were performed using Statistica 13 (TIBCO Software Inc., Palo Alto, CA, USA). The Pearson χ^2^ tests and Fisher’s exact tests were used to test correlations between discrete variables. For survival analysis, Kaplan–Meier estimation with Log-rank test as well as univariate and multivariate regression were applied. All tests were two-sided, and *p*-values < 0.05 were considered statistically significant.

## Figures and Tables

**Figure 1 ijms-25-10274-f001:**
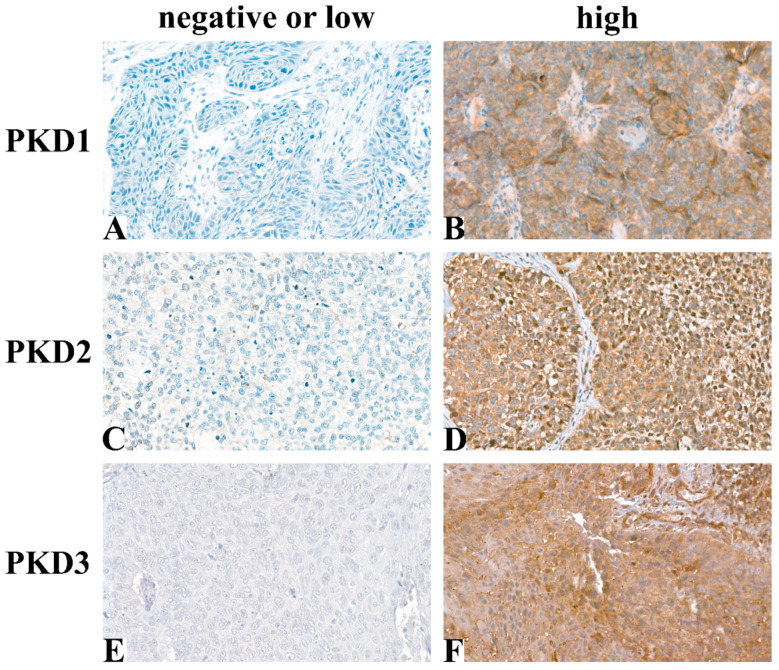
Representative images of immunohistochemical staining: PKD1 low (**A**), PKD1 high (**B**), PKD2 low (**C**), PKD2 high (**D**), PKD3 low (**E**), PKD3 high (**F**) (Magnification: 40×).

**Figure 2 ijms-25-10274-f002:**
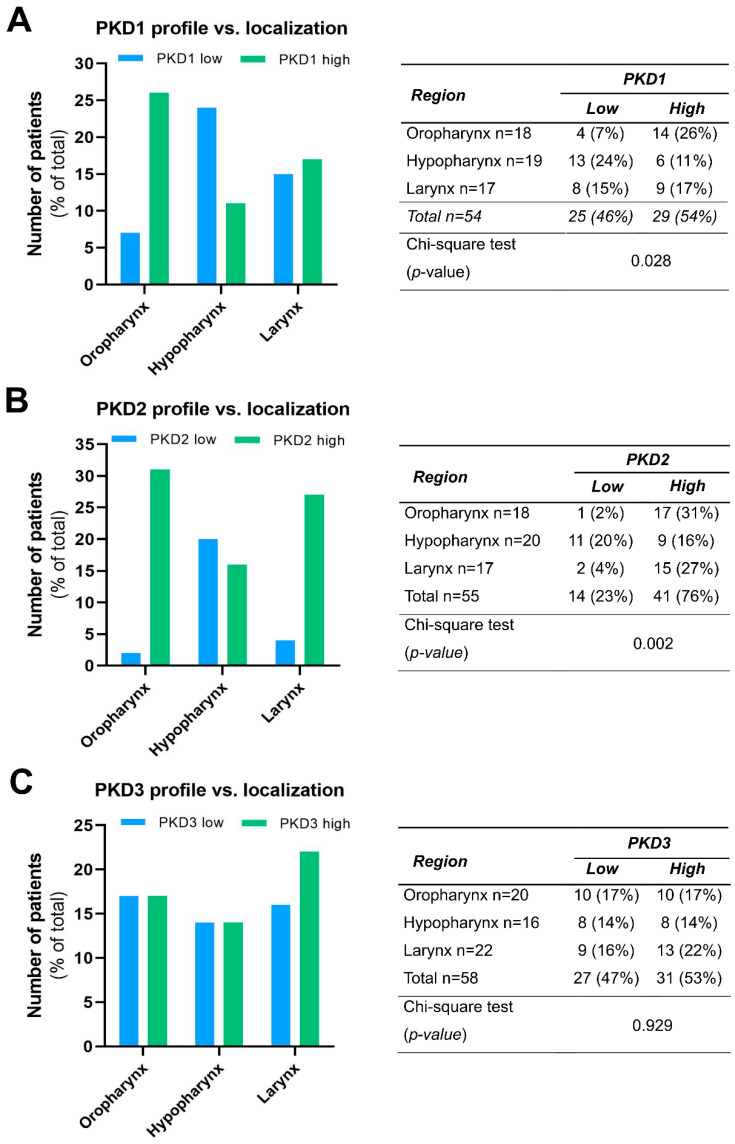
Expression profile of PKD1 (**A**), PKD2 (**B**), and PKD3 (**C**) isoforms in head and neck cancers of different localization. Groups were compared by Pearson χ^2^ test. The graphs and the tables showcase numbers.

**Figure 3 ijms-25-10274-f003:**
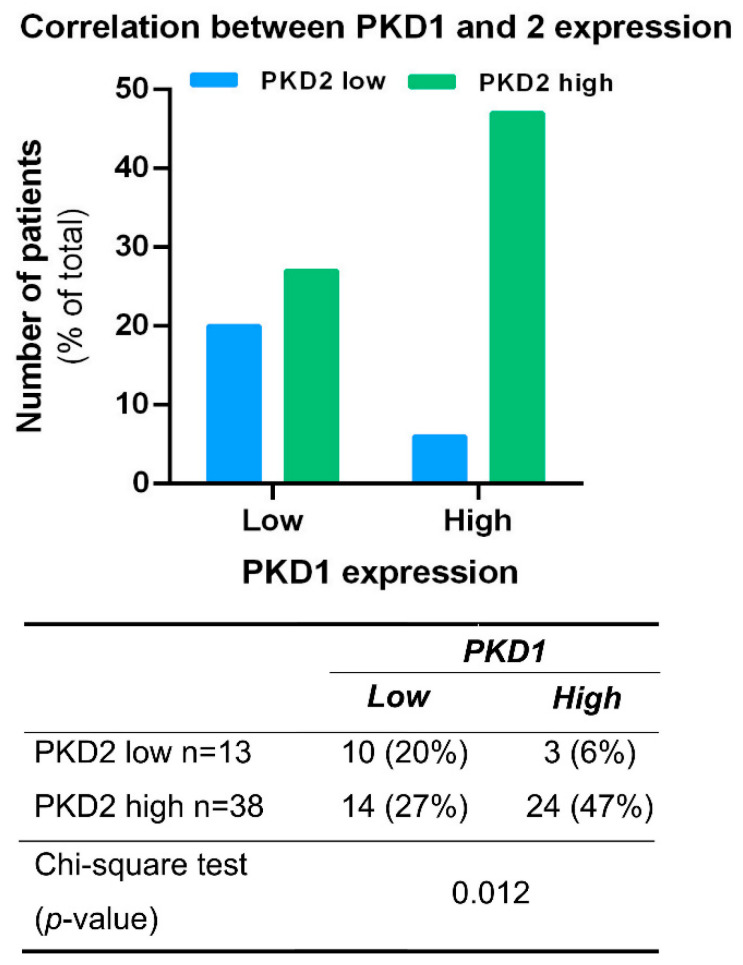
Correlation between PKD1 and PKD2 expression profiles. Groups were compared by Pearson χ^2^ test. The graphs and the tables show the case numbers.

**Figure 4 ijms-25-10274-f004:**
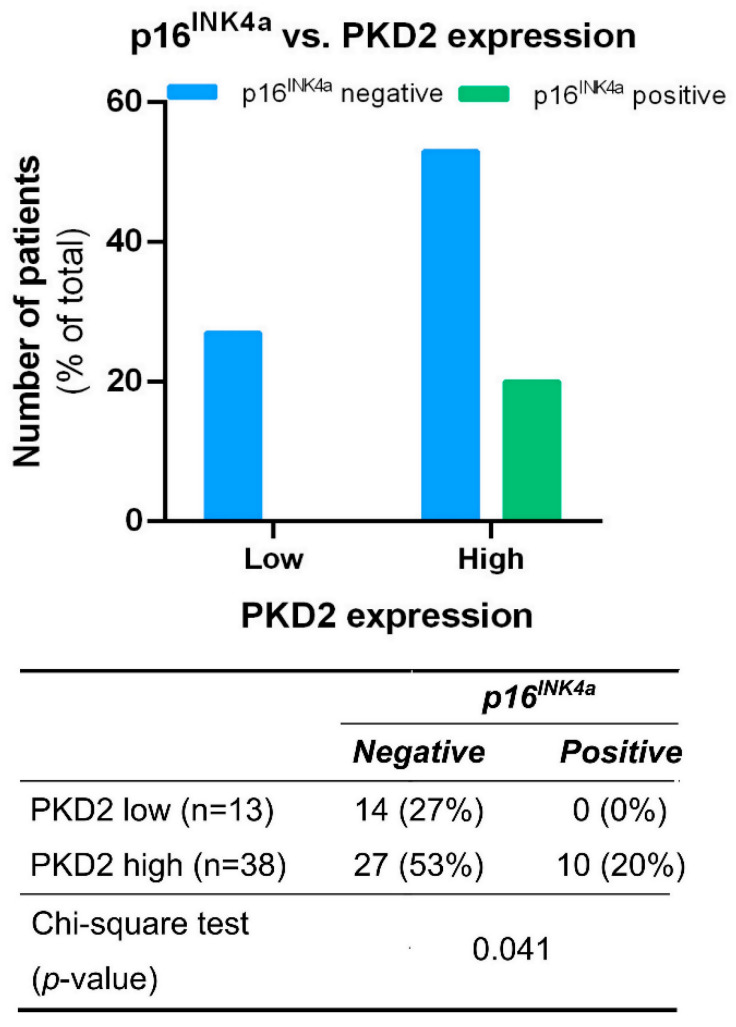
Correlation between PKD2 and p16^INK4a^ protein levels. Groups were compared by Pearson χ^2^ test. The graphs and the tables show the case numbers.

**Figure 5 ijms-25-10274-f005:**
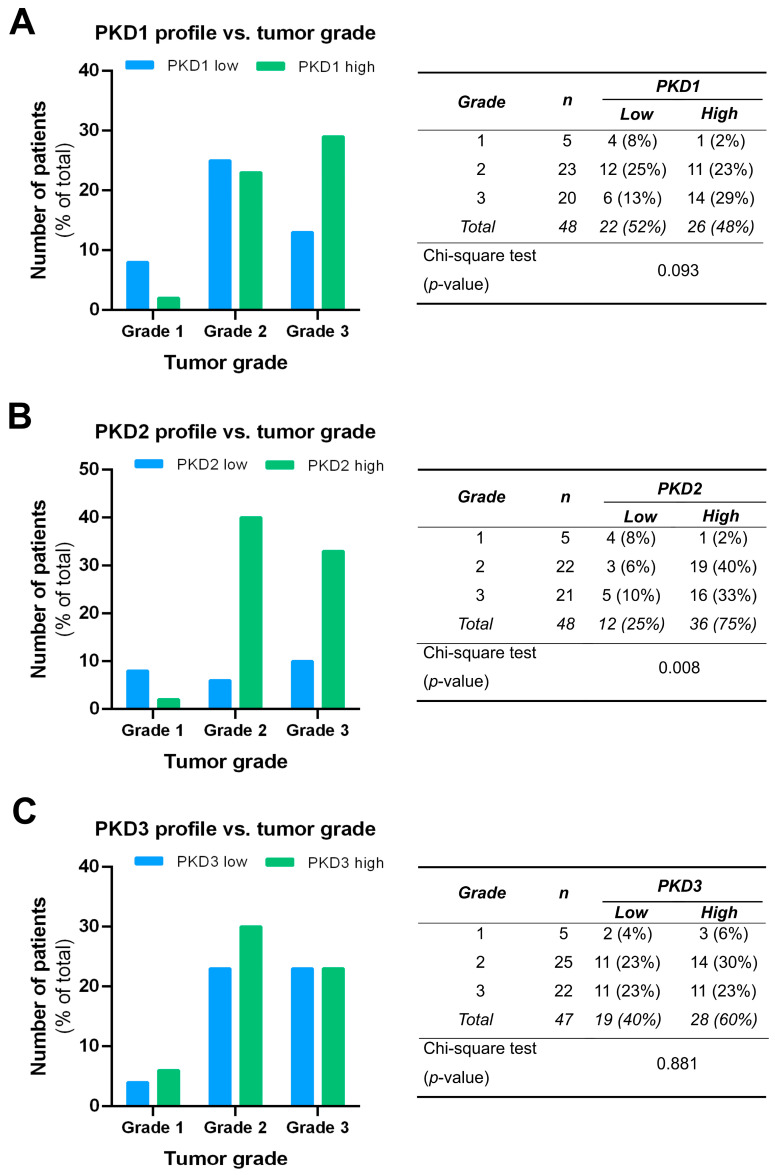
Correlation of tumor grade with PKD1 (**A**), PKD2 (**B**), and PKD3 (**C**) isoform expression. Groups were compared by Pearson χ^2^ test. The graphs and the tables show the case numbers.

**Figure 6 ijms-25-10274-f006:**
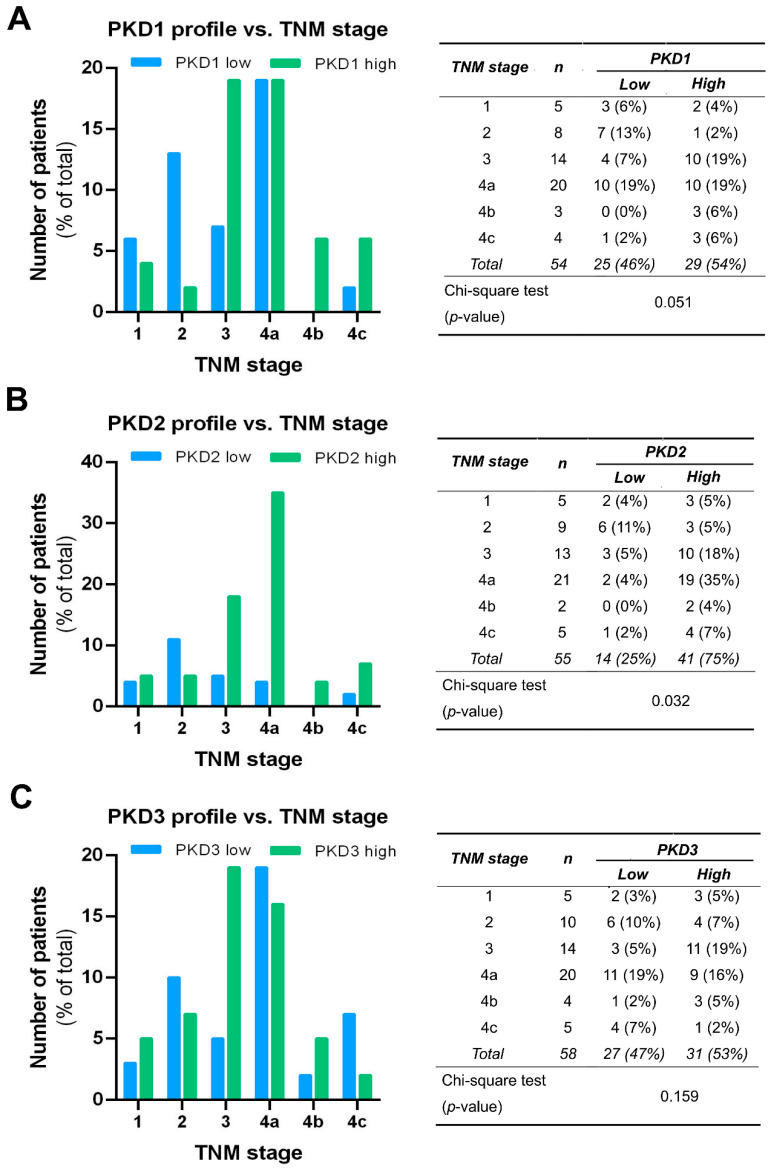
Correlation of tumor, node, and metastasis (TNM) stage with PKD1 (**A**), PKD2 (**B**), and PKD3 (**C**) isoform expression. Groups were compared by Pearson χ^2^ test. The graphs and the tables show the case numbers.

**Figure 7 ijms-25-10274-f007:**
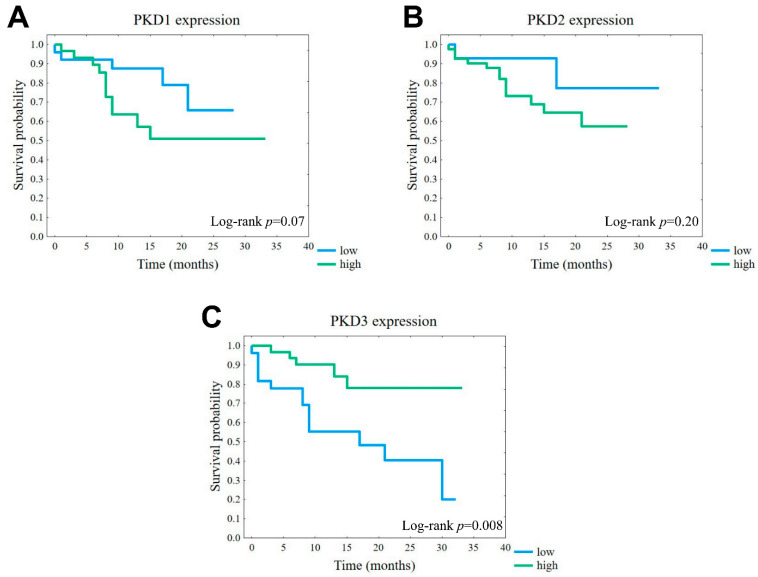
Association of disease-specific survival (DSS) with PKD1 (**A**), PKD2 (**B**), and PKD3 (**C**) isoform expression. Kaplan–Meier plots of overall survival according to the groups of low (blue) or high (green) protein levels.

**Table 1 ijms-25-10274-t001:** Patient characteristics at the time of diagnosis.

Variable	No. of Patients	Variable	No. of Patients
Total no. of patients	63	TNM ^1^ N parameter	
Sex		0	29
Male	53	I	11
Female	10	II A	4
Age (year)		II B	8
Mean	61 (43–79)	II C	8
Tobacco use		III	3
Never	6	TNM ^1^ M parameter	
Previously	18	0	58
Currently	38	I	5
No data	1	TNM ^1^ stage	
Alcohol use		I	5
Never	13	II	10
Previously	23	III	15
Currently	27	IV A	24
Localization		IV B	4
Oropharynx	22	IV C	5
Hypopharynx	22	Grade	
Larynx	19	I	5
TNM ^1^ T parameter		II	29
I	8	III	22
II	19	No data	7
III	23	p16^INK4a^ expression ^2^	
IV A	11	Negative	47
IV B	2	Positive	11
		No data	5
		HPV infection ^2^	
		Negative	52
		Positive	11

^1^ TNM: tumor, node, and metastasis, UICC TNM 7th edition. ^2^ The data of p16^INK4a^ immunohistochemistry and HPV genotyping were analyzed in a previous study [8].

**Table 2 ijms-25-10274-t002:** Proportion of tumors according to biomarker groups.

Variable	No. of Patients
PKD1 all groups	
Negative	6
Low	19
Moderately low	19
Moderately high	8
High	2
No data	9
PKD1 two groups	
Low	25
High	29
No data	9
PKD2 all groups	
Negative	1
Low	13
Moderately low	13
Moderately high	17
High	11
No data	8
PKD2 two groups	
Low	14
High	41
No data	8
PKD3 all groups	
Negative	5
Low	22
Moderately low	17
Moderately high	10
High	4
No data	5
PKD3 two groups	
Low	27
High	31
No data	5

## Data Availability

The data presented in this study are available at the request of the corresponding author for ethical reasons.

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
