# Peer review of "Potential Prognostic Role of Protein Kinase D Isoforms in Head and Neck Cancers"

_ijms, 2024, doi:10.3390/ijms251910274_

Round 1

Reviewer 1 Report

Comments and Suggestions for Authors

1.      In the introduction of the manuscript, a little information about protein kinases and their importance in cancer therapeutics must be added. https://doi.org/10.1016/j.drudis.2023.103852; https://doi.org/10.1016/j.biopha.2024.116332

2.      Altogether 63 therapy naive patients were consecutively enrolled who were diagnosed with squamous cell carcinoma of the oropharynx, hypopharynx. My concern is with the sample size of the study as it is way small. The results are promising but would been more promising if the sample size was much higher.

3.      Another point is the variation in number of males and females taken in the study. The authors could have interpreted the results in terms of sex also to check if they get any trends depending on the sex of patient (male/female)

4.      Limitation of the study must be highlighted.

5.      English terminology at many places seems to have used long sentences and must be improved.

Comments on the Quality of English Language

English terminology at many places seems to have used long sentences and must be improved.

Reviewer 2 Report

Comments and Suggestions for Authors

Overall, their studies were excellently designed. Their data correction and results presentations are acceptable. Their conclusions were based on their results. Based on their results, they concluded that their findings suggest the need to examine protein expression patterns of PKD isotypes using immunohistochemistry as part of future diagnosis and for deducing HNSCC tumor progression and prognosis in such patients. The article is nicely written and  not difficult to follow.  The work is novel and the conclusions are relevant and their manuscript could attract readers as well as advance the field. For most part, the manuscript meet the scientific standard of this journal. Therefore, we should encourage the authors to address some concerns below.

Concerns:

11. The authors examined the patient’s TMA slides for protein expression of the various PDK isoforms using IHC method in which they  immunostained PKD subtype with commercially available specific rabbit polyclonal anti-bodies anti-PKD1, PDK2 and PDK3 respectively.  Though their results are excellent and enabled them to reach their vital conclusions these polyclonal antibodies do not allow us to ascertain whether the detected PKD isoforms are inactive or active forms. This point is very important because active forms of various PKD isoforms are characterized by their specific phosphoprotein forms and there are reliable commercially available monoclonal antibody for each of these three PDK isoforms. For example, there is specific antibody that recognizes active PDK1 as phosphoPDK1Ser240 or recognizes active PDK2 as phosphoPDK2Thr160 and a specific one for active PDK.  Such phosphorylation site specific monoclonal antibodies could have been useful in determining whether in addition to changes in protein expression levels in various TMAs, the increases in protein expression is associated with enhanced activated forms of the various PDKs in various locations of the patient’s TMA they examines in this study.

Did the authors try to use these specific monoclonal antibodies in their IHC analyses of the HNSCC TMAs? If they did, were their results similar to what they saw with polyclonal antibodies?

22.  After each of the various PDKs are activated, it transduces signal along specific signaling pathways by targeting its immediate downstream signaling molecules.  I am wondering whether the authors detected any of the respective immediate downstream targets of PDK1 or PDK2 or PDK3? If they detected them, did their expression/activated levels follow the same pattern as they saw with the PDK protein levels and/or correlated with disease free survival as seen your figures 7?

33.  In some cases, the authors used 54 or 55 or 58 TMAs for the IHC analysis of PDK protein expression instead of using all the samples from all the 63 HNSCC patients. However, they did not explain why they did not use all the samples. A brief explanation in the materials and methods section will be important.

44.  Under each of the figures, they showed the bar graphs and provided the p-values without showing the standard deviations in the graphs. Were the bar graphs plotted and shown under the figures obtained from analysis of a single representative TMA sample or the bar graphs represent averages of all the samples analyzed for each PKD isoform as indicated in the accompanying tables?  If the bar graphs represent averages of the total samples analyzed, then one would expect to see the deviations on the graphs. Please, clarify this point under the various figure legends.

55.  Each PDK isotype can be activated differentially by different molecules and perhaps by various drugs. Do the authors know the types of medication(s) the patients were on prior to taking their tumor samples for the TMAs?

Comments on the Quality of English Language

The quality of English language used in the manuscript is fine and may require only minor editing.

Reviewer 3 Report

Comments and Suggestions for Authors

The article presents an interesting and potentially significant study on the expression of Protein Kinase D (PKD) isoforms in head and neck squamous cell carcinomas (HNSCC). The research highlights how the analysis of the expression of these isoforms could have relevant diagnostic and prognostic implications, particularly in a tumor type with poor established prognostic markers.

However, although the work describes correlations between the expression of PKD isoforms and several clinicopathological parameters, an in-depth analysis of the underlying molecular mechanisms is lacking. A more detailed discussion on how exactly PKD isoforms influence tumor progression and prognosis in HNSCC patients would be useful. Furthermore, the sample of 63 patients may not be sufficient to draw definitive conclusions, especially when looking for associations between molecular markers and clinical parameters. A larger sample size could increase the statistical robustness of the results. Therefore, I would suggest an expansion of the sample. And it is necessary to investigate further how PKD isoforms regulate key processes such as epithelial-mesenchymal transition (EMT), cell proliferation and migration. In vitro experiments could provide more details on these processes.

Round 2

Reviewer 2 Report

Comments and Suggestions for Authors

Thank you for addressing the concerns of the reviewers and for incorporating some your responses into the revised manuscript

Reviewer 3 Report

Comments and Suggestions for Authors

The manuscript has been suitably improved and can be accepted for publication in its present form.